# Animal Models as a Tool to Design Therapeutical Strategies for CMT-like Hereditary Neuropathies

**DOI:** 10.3390/brainsci11091237

**Published:** 2021-09-18

**Authors:** Luca Bosco, Yuri Matteo Falzone, Stefano Carlo Previtali

**Affiliations:** Institute of Experimental Neurology (INSPE), Division of Neuroscience, IRCCS San Raffaele Scientific Institute, 20132 Milan, Italy; bosco.luca@hsr.it (L.B.); falzone.yuri@hsr.it (Y.M.F.)

**Keywords:** animal model, CMT, IPN, pathogenesis, therapy

## Abstract

Since ancient times, animal models have provided fundamental information in medical knowledge. This also applies for discoveries in the field of inherited peripheral neuropathies (IPNs), where they have been instrumental for our understanding of nerve development, pathogenesis of neuropathy, molecules and pathways involved and to design potential therapies. In this review, we briefly describe how animal models have been used in ancient medicine until the use of rodents as the prevalent model in present times. We then travel along different examples of how rodents have been used to improve our understanding of IPNs. We do not intend to describe all discoveries and animal models developed for IPNs, but just to touch on a few arbitrary and paradigmatic examples, taken from our direct experience or from literature. The idea is to show how strategies have been developed to finally arrive to possible treatments for IPNs.

## 1. Introduction and Historical Use of Animal Models in Medicine

Through the course of history, experimentation on non-human vertebrates (hereinafter referred as “animals”) played a pivotal role in the development of biomedical research. At the same time, it also constituted a topic of public debates, raising criticism and controversies of both philosophical and moral nature, depending on the ever-changing perspectives of different ages.

Although the earliest evidence of cranial surgery performed on an animal dates back to the late Neolithic, Ancient Greeks were probably the first to systematically experiment on dissecting animals for anatomical studies [1,2]. During the fifth century BC Alcmaeon of Croton was the first to pioneer the field of comparative anatomy, followed by eclectic personalities such as Aristotle, Herophilus, Erasistratus and Galen, whose works, despite the inevitable anatomical mistakes, remained canonical until the Renaissance [3,4,5].

Notwithstanding the misconception of a period of scientific decadence, animal experimentation never really fell into disuse even during the Middle Ages, carried on by the likes of Leonardo Da Vinci for anatomic discovery. It is during the seventeenth century however that physiology studies marked the dawn of modern scientific research in biomedical sciences, when William Harvey first depicted an accurate description of the circulatory system through the examination of the heart function in eels and several other fishes.

Along with the advances in understanding physiology and pathology, criticisms regarding the use of animals in science emerged, concerning both the validity of conclusions derived from experiments on suffering or dead animals, and the moral question, risen even stronger in light of Darwin’s findings on evolution, which made differences between animals and humans progressively more nuanced [6].

By the beginning of the nineteenth century, scientists such as Francois Magendie and Claude Bernard in physiology, and Louis Pasteur in microbiology, made the use of animal experimentations pivotal towards the validation of the scientific method [7]. In the first half of the 1900s, the discovery and testing of insulin in diabetic dogs, and the development, appraisal and production of a vaccine for poliomyelitis, while requiring the sacrifice of large numbers of monkeys, contributed to saving of millions of human lives [8,9].

These breakthroughs in medical sciences progressively disproved the argument that no medical progress could be obtained through animal research, and the debate shifted towards the need for a regulated environment, aimed at using laboratory subjects in a much more humane, limited and scientifically productive manner [10].

Through the course of the twentieth century, the emergence of rodent species as the preferred laboratory subjects allowed further advancement in biomedical research, due to the multiple additional benefits of such models, including their physiological similarities with humans, their small size, the ease in maintenance, short life cycle and abundant offspring [11].

In 1921 Clarence Cook Little inbred the mouse strain C57BL/6 or “black 6”, which later became the most popular laboratory mouse to date, and whose complete genome was sequenced in 2002 [12].

In 1976 Rudolf Jaenisch proved that foreign DNA could be integrated into the DNA of early mouse embryos through the use of a retrovirus, developing the first transgenic mammals in history, and the development of the first knockout mouse in 1987, granted Mario R. Capecchi, Martin J. Evans, and Oliver Smithies the winning of the Noble Prize in 2007 [13,14]. With the development of the Cre-LoxP and CRISPR-Cas9 recombination technologies, a new era of genome editing sciences began, opening countless possibilities for the understanding of gene function and their influence in several genetic and non-genetic diseases [15,16].

In the last 20 years we witnessed the rise of a large body of literature on the subject, prospects being that it will continue to play a central role in the development of biomedicine in the foreseeable future.

## 2. Brief Introduction to Human Inherited Peripheral Neuropathies (IPNs)

Inherited peripheral neuropathies (IPNs) comprise a vast and heterogeneous group of disorders of the peripheral nerve ranging from pure motor (hereditary motor neuropathy, HMN) to pure sensory (hereditary sensory, HSN, or hereditary sensory-autonomic neuropathy, HSAN), and including the most frequent sensory-motor Charcot-Marie-Tooth (CMT) disease; for recent review see [17,18,19]. First described as nosological entities by Charcot and Marie in France 1886, and almost independently by Tooth in England at around the same time, they have estimated prevalence ranging from 1 in 8500 to 1 in 1200 [18,20,21].

As described by its discoverers, the disease has a typical onset during late childhood or adolescence, although later onset has been also reported, as for example in axonal *myelin protein zero* (*MPZ*) mutations [22,23].

The disease is characterized by distal motor and/or sensory deficits at the four limbs, muscle atrophy, reduced or absent deep tendon reflexes and bone deformities (typically *pes cavus* and hammertoes). Symptoms slowly progress, with peroneal muscles usually bearing the first signs of the disease, leading the foot to drop during walking or running, and the ankle to become unstable, which can lead to traumatic injury. Later, patients may develop atrophy of the hands and forearms, but the wasting rarely extends proximally to the elbow or above the middle third of the thigh, giving the lower limbs the typical “inverted champagne bottle” or stork appearance. Depending on magnitude of sensory and autonomic involvement, patients seldom develop trophic changes of the skin and bones in the affected limbs, due to repeated injury on analgesic parts and lack of autonomic vascular reflexes [24]. In most cases, despite a variable disability course, life expectancy is not reduced.

For a long time, CMTs have been divided in two groups according to nerve conduction velocities, which are typically slow in type 1 (CMT1; mean motor nerve conduction velocities, MNCV < 38 m/s for upper limb studies) and near normal in CMT2, referred as demyelinating and axonal types, respectively [25]. Due to the great heterogeneity in phenotypes, a broad overlap exists between clinical presentations of CMT1 and CMT2, and thus an intermediate form has been introduced (MNCV >35 and ≤45 m/s).

In recent years, new genetic findings enriched the rapidly evolving landscape of hereditary neuropathies and allowed for a better characterization of the clinical peculiarities in different forms, simplifying in some way the matter of their classification. From the first locus mapped on chromosome 1q22–q23 in 1982 [26], myriads of genes have been attributed to CMTs in the last twenty years, thanks to the development of next generation sequencing tools and genome wide analysis [27]. However, many patients still lack a genetic diagnosis in IPNs, primarily for CMT2, HMN and HSN forms.

The molecular mechanisms at the basis of the pathogenesis of all these many forms of CMTs are therefore multiple and involving almost all the biological functions of a cell [28,29]. In demyelinating CMTs, the original dysfunction resides, or prevails, in Schwann cells leading to abnormal nerve development in the most severe (congenital) cases or to the progressive inability to maintain the myelin sheath in later forms. From the initial findings where most of the causative genes were encoding for glia-specific proteins such as myelin components (*PMP22*, *MPZ*), transcription factors (*EGR2*, *SOX10*) or adhesion molecules (*GJB1/CX32*, *PRX*), it was then clear that the causative genes could perturb almost any aspect of the cell function [28,29]. In fact, there are genes involved in energy production, vesicular/membrane trafficking, cytoskeleton rearrangement and cell signaling [29,30]. Thus, any aspect that affects myelin assembly/maintenance, cells survival or the interaction with surrounding structures, primarily axons, would be responsible for demyelination.

Similarly, axonal CMTs mainly rely on primary deficits in the axon/neuronal compartment. In this case, the causative mechanism mainly relies on the long extension of the axon, which in many cases it can extend over one meter. Any (genetic) event affecting/limiting the survival of this distal portion of the neuron would result in axonal damage/degeneration in so called CMT2 neuropathies [31,32]. Accordingly, causative genes described in CMT2 encode for proteins involved in energy production (mainly for mitochondria function), axonal transport, neuronal survival and synaptic transmission. As a consequence, vital molecules and organelles cannot reach nerve terminals (anterograde) or being removed from the periphery to the neuronal soma (retrograde transport), altering neuronal homeostasis and making axons vulnerable to damage [31,32].

Despite the considerable progresses in diagnostic accuracy and molecular classification, there are currently no disease-modifying therapies for this group of genetic disorders. In recent years we have seen the development and commercialization of many gene-based therapies aimed at changing the natural history of other neuromuscular diseases, providing new hope for the development of specific treatments for hereditary neuropathies, as well.

## 3. The Importance of Animal Models in Inherited Neuropathies

The development of animal models has been fundamental to study the pathogenesis of IPNs. If we narrow it down to rodents (and primarily mice), which are the most useful models in the field, they not only reproduced (in most of the cases) clinical and pathological aspects of the human neuropathy, but they also shed light on the molecular mechanisms sustaining the disease and opened routes for therapeutical approaches (Figure 1). A list of the rodents developed to mimic IPNs is presented in Table 1.

### 3.1. Can Animal Models Reproduce Human IPNs?

The most obvious use of animal model is to reproduce human disease. For IPN, the prototype is the rat model of CMT1A, the most frequent form of IPN due to copy number variation of the *PMP22* gene [135,136]. CMT1A patients have three copies of the *PMP22* gene, due to a genomic duplication spanning around 1.5Mb on chromosome 17 and resulting from unequal meiotic crossover that is mediated by highly homologous repeat sequences flanking the duplicated region. Transgenic rat models overexpressing *Pmp22* were generated representing the first direct proof that in humans is the gene dosage of *PMP22* responsible for the disease [137] Accordingly, a rat expressing the equivalent of three genetic copies of the *Pmp22* gene displayed weakness and gait abnormalities, reduced nerve conduction velocities, and peripheral neuropathy with hypomyelination and onion bulbs. Axonal loss was also observed in subsequent studies [48,138]. Increased overexpression of *Pmp22* transcript (obtained by breeding rats in homozygosity) correlates with a dramatic worsening of the disease [139]. Further CMT1A models have been generated in mice, though they needed several extra copies of the *Pmp22* gene to induce overexpression of the Pmp22 protein and thus the neuropathy [36,37,45,140].

A subsequent demonstration that the neuropathy is the direct consequence of *Pmp22* overexpression and could be rescued by reducing *Pmp22* message was shown with the use of an inducible (tetracycline-controlled transcriptional activation) mouse model, in which tetracycline administration was able to switch off *Pmp22* overexpression. In this transgenic mouse, tetracycline administration in adult animals, when *Pmp22* overexpression had already caused the peripheral neuropathy, was able to ameliorate the disease with a partial correction of hypomyelination [33].

The genetic counterpart of CMT1A is the hereditary neuropathy with liability to pressure palsies (HNPP), in which patients present lower *PMP22* copy number (one single allele) and episodic motor and/or sensory deficits [141]. The pathological hallmark of the disease is the presence of focal “sausage-shaped” swellings of the myelin defined tomacula, which are reproduced in mice hemizygous for the *Pmp22* gene [142,143]. Similarly, experimental nerve compression in these mutant mice reproduces nerve conduction blocks by neurophysiological analysis [144].

Finally, CMT1E, which is the consequence of *PMP22* point mutation, is also reproduced in spontaneous *Trembler* and *Trembler-J* mice, both carrying point mutation (respectively Gly160Asp and Leu16Pro) in the *Pmp22* gene also found in humans [65,145]. Additionally, these mutants reproduce clinical and pathological findings of human neuropathies (including onion bulbs), whereas the exact pathomechanism remains unclear. Several findings show that the mutated protein forms aggresomes and is retained in the endoplasmic reticulum (ER) [146,147,148,149,150,151,152], eventually triggering an unfolded protein response (UPR) and Schwann cell damage [153] and/or abnormal calcium entry by dysregulation of store operated calcium channel activity [154].

The pathological hallmarks of all the above *PMP22* models have been extensively described and compared to human in a recent review article [138]. Overall, rodent models of CMT1A, CMT1E and HNPP, but in general for many of the demyelinating CMTs (see sections below), were able to reproduce most of the clinical and pathological findings of human patients and have been instrumental to elucidate their pathogenetic mechanism. Conversely, this was not the case for axonal CMTs, which minimally reproduced human finding but were anyway useful to reveal their pathogenesis in many cases. These aspects are treated and discussed in the following sections.

### 3.2. Pathogenesis of IPN Due to Loss- or Gain-of-Function Mechanism

Animal models have been extremely useful also to discriminate between loss- or gain-of-function mechanism in the pathogenesis of IPNs. The clear example is CMT1B, the second most frequent form of CMT due to heterozygous mutations in the *MPZ* gene. Myelin protein zero (P0) is the glycoprotein encoded by *MPZ* and the most abundant myelin protein in peripheral nerve [155]. P0 has an immunoglobulin-like structure that assembles to form tetramers emanating from the membrane surface, which interact with tetramers (in *trans*) on the opposing membrane surface (of the ensheathing Schwann cell) in order to promote myelin sheath compaction [156].

It was quite obvious to postulate that heterozygous mutations in *MPZ* would have reduced the production of P0, and thus of the component necessary to build and compact myelin, causing the neuropathy with a loss-of-function mechanism, as also recently reported [157]. Accordingly, mice with *Mpz* disruption by homologous recombination showed neuropathy with hypomyelination in heterozygous mutants [59], which is aggravated in homozygosity and undergoes distal axonal loss in older mice [158].

However, the majority of *MPZ* mutations do not cause mild phenotypes as reproduced by heterozygous *Mpz* mice or observed in humans with predicted premature termination and nonsense mediated decay of the mutant allele [23,159,160,161,162,163]. Thus, to definitely prove that most of *MPZ* mutations could act through a gain-of-function mechanism, transgenic mice bearing specific *MPZ* human mutation *S63del* or *S63C* have been generated. In these mice, an extra copy of the mutated allele (*S63del* or *S63C*) was expressed together with two copies of normal *Mpz* alleles [55]. If the mutated allele had acted through a loss-of-function mechanism, one would have expected no phenotype, as two normal *Mpz* alleles were present. Conversely, mice exhibited a typical demyelinating neuropathy, including clinical, neurophysiological and histological phenotype, confirming a typical gain-of-function mechanism [55].

Interestingly, animal models also revealed that mutated P0 glycoprotein may act with gain-of-function mechanism by reaching its target site, myelin, or as a consequence of its retention in the ER. For example, transgenic mice expressing a myc-tag at the mature N-terminus of P0 clearly showed that the (artificially) mutated protein arrives in myelin and disrupts the myelin compaction, possibly by the physical displacement of tetramers in myelin sheath [164]. More intriguingly, mice bearing only the *S63del* mutation showed that the mutated P0 protein does not reach the myelin sheath but is retained in the ER where it elicits a UPR [55]. The UPR is a mechanism generated by cells to respond to stress by activating the transcription of chaperons, attenuate protein translation and/or stimulate protein degradation to eventually reduce the load of improperly folded protein. However, the persistence of this condition becomes maladaptive promoting cell dysfunction or apoptosis. Many evidence showed that attenuation of protein translation is sufficient to ameliorate these forms of peripheral neuropathies, providing a new therapeutic strategy [56,57,165,166,167,168]. Accordingly, Sephin-1, a synthesized drug that prolongs protein translation attenuation, prevented the myelin and motor defects in *P0-S63del* mice [58].

Finally, the generation of a different transgenic mouse, expressing the *P0-S63del* and the wild type P0 with a myc epitope tag at the C terminus showed that the mutated P0 can interfere with the transit of the wt-P0 and reduce its amount in myelin with a direct dominant negative mechanism other than UPR [169].

### 3.3. One Gene but Different Phenotypes

Mutations in CMT genes may also present with a widely heterogeneous phenotype. Sometimes, this variability is caused by mutations in one single gene, as it is the case for *MPZ*. Mutations in *MPZ* may range from very severe forms defined as Dejerine-Sottas or with very early onset as congenital hypomyelination, to classical demyelinating CMT with moderate to mild phenotype, axonal CMT or intermediate CMT [170]. In the previous section, we already discussed as mouse models could mimic either milder demyelinating forms with heterozygous *Mpz* (+/−; loss of function) mice, or more classical moderate to severe demyelinating forms with *S63del* and *S63C* (gain of function) mutant mice respectively [55,59]; with the note that in humans, the neuropathy due to S63C is more severe than *S63del*, while in mice it is the reverse. Interestingly, simple overexpression of normal wild type P0 protein reproduced congenital hypomyelination phenotype [171]. This is partly due to altered trafficking of P0 in promyelin-forming Schwann cells, where the abnormal expression of a “sticky” Ig-like protein such as P0 impairs/arrests axonal sorting and axon ensheathing during development resulting in several Schwann cells arrested in a 1:1 relationship with axons, a typical congenital hypomyelination aspect.

Moreover, a recently developed mouse model carrying the *MPZ-T124M* mutation reproduces the axonal phenotype of human CMT2J/I (Maurizio D’Antonio and Shackleford Ghjuvan personal communication and presentation at PNS Society meeting 2021). We may predict that this mouse model will likely reveal the pathomechanism of axonal damage caused by some *MPZ* mutation, which is still obscure.

### 3.4. Cell Autonomy in the Pathogenesis of IPN

Animal models have been also fundamental to reveal cell autonomy in the pathogenesis of some CMT forms. In fact, and differently than initially though, many genes mutated in CMT do not encode for specific nerve proteins but for ubiquitously expressed proteins [172]. Whether loss/abnormal function of some proteins could affect both Schwann cells and neurons (axons), or only one of these two compartments was to be demonstrated. In this respect, the use of conditional mouse mutants by means of the Cre/LoxP technology became essential. With this system, the gene of interest to be ablated is flanked by LoxP sites, which are recognized by the Cre-recombinase enzyme expressed by specific transgene only in the cell of interest [173].

CMT4B1 is an autosomal recessive IPN due to mutations in the *MTMR2* (*myotubularin-related 2*) gene, encoding for a phosphoinositide -PtdInsP3 and PtdIns(3,5)P2- phosphatase, characterized by early onset severe polyneuropathy [174]. The disease is characterized by demyelinating and axonal features and typical excessive redundant myelin outfoldings at nerve pathology [175]. Homozygous deletion of the *Mtmr2* gene in null mice reproduced the typical myelin outfoldings phenotype [109,110]. Whether myelin outfoldings were generated by a loss of Mtmr2 in Schwann cells or neurons was solved by conditional mouse mutants. Mutants with Schwann cell conditional deletion of *Mtmr2*, obtained by *P0-Cre* transgene expressed selectively in Schwann cells [176] revealed nerve pathology and myelin outfoldings identical to the full KO mouse [177]. Conversely, conditional deletion of *Mtmr2* in motoneurons, with the *HB9* transgene [178], did not result in any overt phenotype suggesting that loss of Mtmr2 in Schwann cells, but not in motor neurons, is both sufficient and necessary to cause CMT4B1 neuropathy [177].

Conversely, conditional mouse models showed that CMT4J is due to defective Fig4 function in both Schwann cells and neurons. Fig4 is a 5-phosphatase involved in the dephosphorylation of PtdIns(3,5)P2. CMT4J is a severe autosomal recessive demyelinating neuropathy with childhood onset [117,179]. When *Fig4* was ablated in neurons or motoneurons, respectively by *Syn1-Cre* or *HB9-Cre* transgene, mice developed progressive neuronal and axonal degeneration [180,181]. Similarly, when *Fig4* was ablated in Schwann cells, by *P0-Cre* transgene, these cells showed impairment in endosomal trafficking and autophagy resulting in demyelination [181]. Thus, both neurons and Schwann cells need FIG4 to sustain proper nerve function.

Interestingly, as both Mtmr2 and Fig4 have different effects on PtdIns(3,5)P2, respectively; Fig4 has a role in the generation of PtdIns(3,5)P2 and Mtmr2 in its dephosphorylation—double mutant mice showed that this could be a therapeutic target for CMT4B. In fact, reduction of Fig4 in *Mtmr2−/−;Fig4+/−* mice ameliorated the myelin phenotype [182]. Finally, replacement of Fig4 in neuronal compartment in full *Fig4* KO mice was sufficient to rescue mouse survival [180].

### 3.5. Ex Vivo Models of IPN

Animal models may also facilitate in reproducing in vitro models of the disease, which may reveal further knowledge on the molecular mechanisms that sustain the IPN. One clear example comes from dorsal root ganglia (DRG) explants from the *Mtmr2* KO mouse model of CMT4B1. Sensory DRG neurons and Schwann cells can be plated in vitro, either as isolated cells or as a whole in the entire DRG, and upon conditioned media and ascorbic acid they generate myelin segments; recently reviewed in [183]. As mentioned above, CMT4B1 neuropathy is characterized by myelin outfoldings at nerve pathology in both human and mouse models. Strikingly, DRG explants from both full KO and Schwann cell conditional KO mouse reproduced similar findings in vitro [184], and this in vitro model was used to reveal possible interactors and to test potential rescue therapies [111,182,185]. Moreover, this was applied to other mouse models, such as *Mtmr13* KO, also reproducing myelin outfoldings typical of CMT4B2 [186].

Another example of in vitro model of IPN was achieved with small *heat-shock protein B1* (*HspB1*) mutant mice reproducing axonal CMT2F and HMN 2B [187,188]. HSPB1 is a ubiquitously expressed chaperone protein preventing aggregation of misfolded or non-native proteins and regulating several events including cytoskeleton stabilization, neurofilament assembly, apoptosis and autophagy [189]. Mutations in heat shock proteins alter their binding affinity to client proteins, in this case for HSPB1 regulating acetylation of α-tubulin in microtubules of peripheral nerve, as shown in the specific mouse mutants and in their DRG neurons explants [95,190]. Microtubule instability and thus deficits in axonal transport was dramatically rescued by drugs inhibiting histone deacetylase 6 (HDAC6) activity as revealed by DRG neurons isolated from symptomatic mutants and then confirmed in mice [95]. As inhibitors of HDAC6 restored axonal transport, including mitochondria transport and (possibly) dynamics, these drugs were then shown to produce benefits also in mouse models of CMT2A [73,191], reproducing the axonal neuropathy characterized by abnormal mitochondria fusion and transport due to mutations in *MFN2* gene [191,192].

### 3.6. Is It Possible to Reproduce Length-Dependent Axonal IPN?

Several other models of IPNs have been generated in the last 20 years, whose description is not the focus of this manuscript, but can be found in Table 1. In many cases, they reproduced most of the clinical and pathological aspects of human neuropathies, whereas in others, namely axonal IPNs, the neuropathy was not induced, or it was very weak.

This was well represented in the case of the *NEFL* gene, whose mutations cause CMT2E and CMT2F [193,194]. These are axonal neuropathies exhibiting a broad range of phenotype ranging from mild to moderate and severe. *NEFL* encodes for the light (molecular weight) neurofilament protein (NFL), which associates with subunits of the medium (NFM) and heavy (NFH) molecular weight to generate coiled-coil dimers to form an antiparallel tetramer and then 10-nm filaments [195]. The mutated NFL acts by disrupting the neurofilament network causing protein aggregation and inclusions, and thus affecting axonal transport [196,197]. Surprisingly, the generation of *Nefl* KO mice did not result in overt phenotype, except for minimal reduction of axon caliber (by 15%), minor abnormality in nerve regeneration after crush [198], and some effect on motor function and spatial orientation revealed by sophisticated analyses [199]. Moreover, no phenotype was also reported in KO mice for *Nfm* and *Nfh* [200,201]. Transgenic mice expressing the p.P22S or the p.E396K *Nefl* mutation and knockin for p.P8R also did not exhibit peripheral neuropathy [90,91,202].

A more consistent phenotype was revealed by *knockin* mice, harboring heterozygous p.N98S *Nefl* mutation [91,92], one of the most common mutations in human. These mutants showed abnormal behavioral and nerve conduction studies consistent with neuropathy, and the histology revealed reduced axonal size and paucity of neurofilaments, a modest reduction of the number of myelinated fibers and very rare degenerating fibers [92]. Anyway, even in this case the phenotype was significantly weaker as compared to humans, and this was also observed for other mutants for axonal CMTs such as *Gdap1*-null mice [98,139], *Hint1* [203,204], *Mfn2* [74,77,78,79,205] and *Mme* [206]. This is likely due to many factors. First, the relatively reduced lifespan of mice, which do not live long enough to develop disorders with a very slow progressive mechanism such as axonal neuropathy.

Second, mouse nerves are too short to manifest a length-dependent axonal loss, and this should be searched in very distal axons as those in the hindpaw [207], in the architecture of neuromuscular junctions [84,98,104], or by stressing the system by mechanical nerve injury to reveal regenerating defects [208]. Finally, the exploitation of ex vivo models, such as DRG/primary neuronal explants from rodents modelling axonal CMT, was regardless very successful to reveal the pathogenetic mechanism of the disease and in reproducing the axonal defect, such as for *HSPB1, MFN2, GDAP1* and others [95,98,191].

### 3.7. Can CMT Models Address Complex Schwann Cell-Axon Interactions?

Common fate of either demyelinating or axonal IPNs is the progressive loss of nerve fibers, which results from axonal degeneration independently of the first origin of the disease (axon or Schwann cell). Whether it is intuitive to understand consequences on Schwann cell function/integrity in case of primary axonal degeneration, the reverse is less obvious. In other words, it is unclear why mutant Schwann cells fail to support axonal function and survival, and it still remains a mostly unanswered question.

Secondary axonal damage is reproduced by rodent models of demyelinating CMTs, with the same limits (mild pathological evidence) described above for pure axonal forms [48,127,158].

It is not clear if dys/demyelination by itself is the cause of (or is sufficient to cause) axonal loss, or other Schwann cell functions intervene to support axon integrity and function. Pharmacologically treated rat model of CMT1A revealed that it is possible to uncouple demyelination from axonal loss [50], suggesting that lack of myelin sheath is not *per se* sufficient to cause axonal degeneration, as similarly described in mouse mutants in which Schwann cells cannot synthesize cholesterol [209,210] or in the *shiverer* mice (involving CNS myelin) [211].

The prevailing theory is that the lack of energy supply by mutant Schwann cells may weaken axonal integrity/function, thus promoting degeneration [210,212,213]. Alternative or complementary factors may include the transfer of proteins or ribosome to axons, rather than the production of toxic factors by mutant Schwann cells [214]. Accordingly, ER stress resulting from accumulation of mutant proteins in Schwann cells, as previously described in different CMT models [215], would compromise the energy support to axons. Further supporting evidence comes from studies in different experimental settings, as for example the recent discovery that Schwann cells shift to glycolytic activity to protect axons and delay degeneration in injured nerve [216]. Similarly, pyruvate supplementation in CMT1E mouse model supports axonal integrity by supplying (mitochondria) energy loss of the distal axons [217].

In summary, animal models are participating in shedding lights on the complex relationship existing between Schwann cells and axons and on the mechanisms of neuroprotection for long-term axonal integrity.

## 4. The Importance of Animal Models to Develop Therapies in Inherited Neuropathies

Finally, animal models have been used to test potential therapies since ancient times. The identification of genes responsible for IPNs, the dissection of possible pathomechanisms and the recognition of involved molecular pathways and target molecules allowed to the development of numerous potential therapeutic strategies in the last 10 years. These therapies include classical drugs consisting of chemical products (new molecules or repurposing of old ones), and innovative therapies consisting of nucleic acids as antisense oligonucleotides (ASO), gene therapy and gene editing.

We will not touch on all therapeutic attempts developed for IPNs, but simply report a few examples of potential therapies under investigation in which animal models have been fundamental.

### 4.1. Conventional Drugs

The identification of the pathogenetic mechanism responsible for specific IPNs, or groups of them, promoted the discovery of new molecules or the repurposing of old ones with the aim to rescue, or ameliorate, clinical and pathological findings of IPNs. Chemical drugs usually have a wider spectrum of use, being useful for different IPN forms with a similar mechanism of action.

Clear examples have been already described above, such as the development of drugs targeting the UPR and tested in mouse models of gain-of-function CMT1B (Mpz) and CMT1E (*Pmp22* point mutation; Trembler J) [58,153,165], or HDAC6 inhibitors for CMT2F (*Hspb1*), CMT2A (*Mfn2*) and CMT1A [38,73,95,218].

A further example of molecule that may be used in different CMTs and revealed by the discovery of the pathomechanism in animal model is Niaspan. Niaspan is an extended-release formulation of nicotinic acid/niacin, used to treat dyslipidemia. Interestingly, this drug was known to modulate (enhance) TACE activity, which is a negative regulator of neuregulin 1 (NRG1) type III and thus myelination in peripheral nerve [219]. We know from seminal studies by Klaus Nave and Jim Salzer labs that NRG1 type III is the key molecule to dictate which axon must be myelinated and the amount of myelination, reviewed in [220]. This was revealed by in vitro and in vivo studies by generating mice with reduced (NRG1 type III +/−) or increased (transgene) expression of NRG1 type III [221,222]. Some CMT subtypes, in particular HNPP and CMT4B1, are characterized by focal hypermyelination such as tomacula and myelin outfoldings, respectively (see above; [109,142]). Proof of principle studies in DRG explants first revealed that downregulation of NRG1 type III was sufficient to significantly reduce tomacula or myelin outfoldings, and then this was confirmed in vivo in mouse models by Niaspam administration (and thus increased TACE activity and consequently reduced NRG1 type III function) [111]. This strategy might represent as a unifying treatment not only for HNPP and CMT4B1 but also for other hypermyelinating neuropathies including CMT4B2, CMT4B3 and CMT4H.

A further interesting drug with possible generalized function in CMTs is Curcumin. This is a polyphenol extracted from a plant with known multiple cellular targets and biological effects including anti-inflammatory and anti-oxidant function, and activity as a low affinity SERCA inhibitor [223]. As SERCA inhibitors can relieve ER stress, it is expected to ameliorate the phenotype in case of UPR activation. Accordingly, curcumin treatment (with different formulations) improved the peripheral neuropathy of *R69C* CMT1B mice, a well-known model of activated UPR response [54], as well as mitigated the neuropathy in *Trembler-J* mice (CMT1E model), which also present ER retention of their mutant protein [224]. However, curcumin presents unfavorable pharmacokinetic. More recently, a new and more stable formulation in nano-crystals showed to have a significant efficacy also in rat models of CMT1A, possibly due to the anti-oxidant effects [225].

### 4.2. Innovative Drugs: ASO, Gene Therapy and Gene Editing

CMT1A is the consequence of the duplication of one allele of PMP22 resulting in patients having 3 copies of the *PMP22* gene (see Section 3.1). Any strategy aimed at reducing the *PMP22* gene dosage (or RNA/protein PMP22 production) would constitute and efficacious strategy to treat the neuropathy. The development of antisense oligonucleotides (ASO) targeting PMP22 expression would constitute an ideal and reliable strategy to treat CMT1A, with the main warning to not exceed in PMP22 repression to avoid the induction of a HNPP-like disease [141]. This strategy was achieved by Zhao et al. by treating two rodent models (mouse and rat) of CMT1A with *Pmp22*-targeting ASO promoting the reduction of 35% of *Pmp22* mRNA [39]. They demonstrated that initiation of treatment after the onset of the disease was able to restore myelination and nerve conduction velocities to levels comparable to normal controls. Moreover, they also revealed expression changes in several molecules after treatment that may become useful biomarkers to monitor the disease and the treatment response. One main limitation is that ASO do not cross sufficiently the blood-brain-barrier, as alternative strategies have been used with success to treat spinal muscle atrophy (SMA; reviewed in [226]). However, at least in rodents, Zhao et al. showed that low levels of ASO are sufficient to cross the blood–nerve-barrier and to reach Schwann cells.

Another innovative strategy is gene therapy. In this case a carrier viral vector is used to deliver a gene, a gene silencer or a gene modifier, into the target host. Recent advances in this strategy for CMT have been reviewed in [30]. Again, rodent models of CMT neuropathies have been instrumental to generate proof-of-principle results. Most interesting, although still with limited translatability, are those relating the intraneural injection of an AAV2/9 vector to silence *Pmp22* and systemic (intrathecal) AAV9 mediated delivery of *Gjb1/Cx32* to treat CMT1X and of *Fig4* gene to treat CMT4J. In the first case, a murine Pmp22-targeting small harpin RNA was delivered into the sciatic nerve of CMT1A rats restoring the expression levels of Pmp22 to wild type condition and promoting nerve myelination and preservation of motor and sensory function of the target limb [49]. In the other cases, lumbar intrathecal injection of the carrier vector resulted in widespread distribution of the carried gene in the peripheral nerve of the mouse model and sustained *Cx32* expression in these nerves, as well as improved motor functions and nerve conduction velocities [122]. Similarly, intracerebroventricular injection of the human *Fig4* gene driven by a ubiquitously expressed chicken β-actin promoter with a CMV enhancer showed increased survival of treated mice, with normal motor functions and amelioration of pathological finding [227]. While intraneural treatment would be difficult to translate in efficient therapy in humans, the other approaches may be favored by the use of a small models as the mouse, and in this case the approach should be revaluated in larger animals to confirm the efficacy of AAV intrathecal/intracerebroventricular administration to reach target cells in peripheral nerves.

Gene editing, using the CRISPR/Cas9 approach, has been tested as a proof-of-concept in animal models of CMT1A. Again, the aim was to reduce the expression of the supernumerary allele(s) responsible for the disease. This was achieved in mouse (C22) model of CMT1A through the direct intraneural delivery of CRISPR/Cas9 designed to target TATA-box of the *Pmp22* gene [40]. Although this an interesting and very promising approach, still the translation in clinical practice is limited by off target activity of the gene editing.

## 5. Conclusions and Warnings

We reported few of many examples of how the use of animal models, particularly rodents, has been useful in the field of CMT neuropathy, ranging from the understanding of developmental processes to the discovery of the pathogenetic mechanisms, till the development of promising future therapies. However, while animal models have been extremely useful to generate this knowledge, we should not forget that their use implies ethical aspects and that not always what is useful in mice or rats it is also working in human. The rule of 3Rs—Replacement, Reduction and Refinement—should constitute the basis for all the scientists working with animal models. Replacement, as the use of other methods whenever possible to replace the use of animals; Reduction, to minimize the number of animals to be used for obtaining a defined result; Refinement, to minimize animal sufferance during experimentation. Moreover, there are cases in which astonishing therapeutic results obtained in animal models (mostly rodents) are not reproduced in humans. Paradigmatic was the use of ascorbic acid to treat CMT1A neuropathy: while treatment in mouse model of CMT1A (C22) significantly ameliorated the neuropathic phenotype and reduced Pmp22 level [41], clinical trials in humans failed to show any effect in patients as well as effect on PMP22 expression [228,229,230,231]. This raises the argument that not always animal models (at least rodents) can reproduce human beings in all its aspects, but also that on occasion, preclinical studies in animal models are not carried out with the same rigor as in humans. In many cases, the number of treated animals is too small, scientists are not blinded to treatment, results are not confirmed to other laboratories, also due to limited funding. This aspect should be better considered by stakeholders as the investments necessary to perform clinical trials in humans are very expansive.

## Figures and Tables

**Figure 1 brainsci-11-01237-f001:**
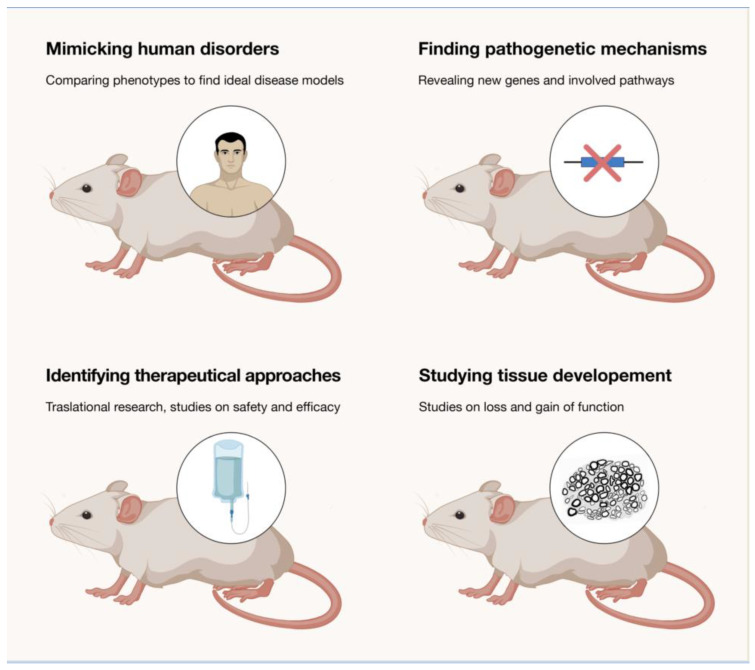
Use of rodent models in IPNs.

**Table 1 brainsci-11-01237-t001:** Overview of rodent models for IPNs.

CMT Subtype	Gene	Model	Mutation	Phenotype (Neuropathy)	Conduction Studies	Nerve Pathology	Experimental Therapy	References
**CMT1A**	PMP22	Transgenic mouse	PMP22 JP18/JY13tg; tTA	Mild	Demyelinating	Mild demyelination	siRNA	[33,34]
**CMT1A**	PMP22	Transgenic mouse	PMP22 duplication (three–four copies) C3-PMP	Mild	Demyelinating	Mild hypomyelination		[35]
**CMT1A**	PMP22	Transgenic mouse	PMP22 duplication (four copies) C61 Het	Normal	Demyelinating	Mild hypomyelination		[36]
**CMT1A**	PMP22	Transgenic mouse	PMP22 duplication (seven–eight copies) C22 Het	Present	Demyelinating	Severe hypo/dysmyelination	Ascorbic acid; geldanamycin; HSP90 inhibitor (NVP-AUT922); ASO treatment; CKD-504-HDAC6 nhibitor; CRISPR/Cas9-mediated downregulation of PMP22 via targeting TATA-box; miR-381; soluble NRG1	[37,38,39,40,41,42,43,44]
**CMT1A**	PMP22	Transgenic mouse	PMP22 duplication (16 copies) TgN248	Present	Demyelinating	Severe hypo/dysmyelination		[45,46]
**CMT1A**	PMP22	Transgenic mouse	PMP22 duplication (unknown copies) MY41	Present	N/A	Severe hypo/dysmyelination		[47]
**CMT1A**	PMP22	Transgenic rat	PMP22 duplication (three copies)—CMT rat	Present	Demyelinating	Severe hypo/dysmyelination and onion bulbs	Progesterone antagonist; AAV2/9; soluble neuregulin1	[44,48,49,50]
**CMT1B**	MPZ	Transgenic mouse	I106L; Mpz+/+	Present	Demyelinating	Severe hypo/dysmyelination and onion bulbs		[51]
**CMT1B**	MPZ	Knock-in mouse	MPZ Q215X/+	Normal	N/A	Mild hypo/demyelination		[52]
**CMT1B**	MPZ	Knock-in mouse	MPZ R98C/+ and R98C/R98C	Mild	Demyelinating	Mild hypo/demyelination	Curcumin derivates	[53,54]
**CMT1B**	MPZ	Transgenic mouse	MPZ S63C	Present	Demyelinating	Mild hypo/demyelination		[55]
**CMT1B**	MPZ	Transgenic mouse	MPZ S63del	Present	Demyelinating	Severe hypo/demyelination and onion bulbs	Chop ablation; sephin 1; over-expression of Nrg1 type-III; inhibition of eIF2α phosphorylation	[54,55,56,57,58]
**CMT1B**	MPZ	Knock-out mouse	MPZ−/−	Present	Demyelinating	Severe hypo/demyelination		[59,60]
**CMT1B**	MPZ	Knock-out mouse	MPZ−/+	Mild	Demyelinating	Mild hypo/demyelination		[60]
**CMT1C**	LITAF	Transgenic mouse	Simple W116G/W116G; Simple W116G/+	Mild	Demyelinating	Mild dysmyelination		[61]
**CMT1C**	LITAF	Knock-in mouse	SimpleT115N/+; SimpleT115N/T115N	Mild	Normal	N/A		[62]
**CMT1D**	EGR2	Knock-out mouse	Egr2+/+ and −/−	Present	N/A	Normal in Het, lethal in Homo		[63]
**CMT1E**	PMP22	Mutant mouse	Exon V deletion (trembler-Ncnp) Het	Present	N/A	Demyelination		[64]
**CMT1E**	PMP22	Mutant mouse	Exon V deletion (trembler-Ncnp) Hom	Present	N/A	Hypomyelination/amyelination		[64]
**CMT1E**	PMP22	Mutant mouse	PMP22 point mutation (G150D) (Trembler) Het	Present	N/A	Severe dysmyelination		[65]
**CMT1E**	PMP22	Mutant mouse	Pmp22 point mutation (H12R) (Trembler-m1H) Het	Present	N/A	Severe hypomyelination and reduced axon number		[66]
**CMT1E**	PMP22	Mutant mouse	Pmp22 point mutation (Y153X) (Trembler-m2H) Het	Present	N/A	Severe hypomyelination and reduced axon number		[66]
**CMT1E**	PMP22	Mutant mouse	PMP22 spontaneous point mutation (L16P) (Trembler-J)	Mild	N/A	Severe hypo/demyelination and onion bulbs	HSP90 inhibitor (NVP-AUT922); Neurotrophin-3; Rapamicyn (in vitro); ACE-083; intermittent fasting; curcumin	[43,65,67,68,69,70,71,72]
**CMT2A**	MFN2	Transgenic mouse	hMFN2 R94QL51/+ (Mitocharc 1)	Mild	Normal	Mild axon atrophy	SW100 (HDAC6 inhibitor); salubrinal	[73,74,75]
**CMT2A**	MFN2	Transgenic mouse	MFN2 R94Q/+	Present	N/A	Mild axon atrophy	transgenic MFN1 overexpression	[76]
**CMT2A**	MFN2	Knock-in mouse	MFN2 R94W/+	Mild	N/A	Normal		[77]
**CMT2A**	MFN2	Knock-in mouse	MFN2 T105M/−	Mild	N/A	Normal		[78]
**CMT2A**	MFN2	Transgenic mouse	MFN2 T105M/+	Mild	N/A	Normal		[79]
**CMT2A**	MFN2	Transgenic mouse	MFN2 T105M/T105M	Present	N/A	Reduced axon number		[79]
**CMT2A**	MFN2	Transgenic mouse	MFN2R94QL87/R94QL87 (Mitocharc 2)	Mild	N/A	Reduced axon number		[74]
**CMT2B1**	LMNA	Knock-in mouse	LMNA R298C/R298C	Normal	Normal	Normal		[80]
**CMT2D**	GARS	Transgenic mouse	AdhGARS G240R	N/A	N/A	Normal		[81]
**CMT2D**	GARS	Mutant mouse	GARS C201R/+	Present	Mixed	Mild axon atrophy	HDAC6 inhibitors	[82,83,84,85]
**CMT2D**	GARS	Mutant mouse	GARS NMF249/+	Present	Mixed	Mild axon atrophy		[86]
**CMT2D**	GARS	Mutant mouse	GARS P234KY/+	Present	N/A	Normal	VEGF; HDAC6 inhibitors	[87,88]
**CMT2D**	GARS	Knock-in mouse	ΔETAQ/+	Present	Mixed	Reduced axon number and axon atrophy	AAV9-mediated allele specific knockdown	[89]
**CMT2E**	NEFL	Transgenic mouse	hNF-LP22S; tTa	Mild	N/A	Normal		[90]
**CMT2E**	NEFL	Transgenic mouse	hNF-LE397K (NF-L−/−)	Mild	Demyelinating	Mild axon atrophy		[91]
**CMT2E**	NEFL	Knock-in mouse	Nefl P8R/+; P8R/P8R	Normal	N/A	Normal		[92]
**CMT2E**	NEFL	Knock-in mouse	NeflN98S/+	Present	N/A	Mild axon atrophy		[92]
**CMT2F**	HSPB1	Transgenic mouse	hHSPB1 R136W	Normal	Axonal	Mild axon atrophy		[93]
**CMT2F**	HSPB1	Transgenic mouse	HSP27-S135F	Mild	Axonal	Reduced axon number		[94]
**CMT2F**	HSPB1	Transgenic mouse	P182L	Mild	Axonal	Reduced axon number		[95]
**CMT2F**	HSPB1	Transgenic mouse	R127W, P182L in ROSA26 locus	Normal	Normal	Normal		[96]
**CMT2F**	HSPB1	Transgenic mouse	S135F	Mild	Axonal	Reduced axon number	HDAC6 inhibitors	[95,97]
**CMT2K**	GDAP	Knock-out mouse	GDAP−/−	Present	Axonal	Reduced axon number		[98]
**CMT2L**	HSPB8	Transgenic mouse	HSPB8 K141N	Present	Axonal	Reduced axon number		[99,100]
**CMT2L**	HSPB8	Knock-in mouse	HSPB8 K141N/+	Normal	Normal	Normal		[101]
**CMT2L**	HSPB8	Knock-in mouse	HSPB8 K141N/K141N	Mild	Axonal	Reduced axon number		[102]
**CMT2O**	DYNC1H1	Knock-in mouse	H304R/+	Mild	N/A	Alteration of NMJ		[103]
**CMT2O**	DYNC1H1	Knock-in mouse	H304R/H304R	Present	N/A	Alteration of NMJ		[104]
**CMT2P**	LRSAM1	Knock-out (gene trap) mouse	RRK239, RRK461Null mutation	Normal	Normal	Reduced axon number		[105]
**CMT2Q**	DHTKD1	Knock-in mouse	DHTKD1 Y486*/Y486*	Mild	Normal	Mild axon atrophy		[106]
**CMT2Q**	DHTKD1	Knock-out mouse	DHTKD1−/−	Present	Mixed	Reduced axon number		[107]
**CMT4A**	GDAP1	Knock-out mouse	GDAP1−/−	Normal	Demyelinating	Normal		[108]
**CMT4B1**	MTMR2	Knock-out mouse	MTMR2−/−	Mild	Demyelinating	Aberrant myelin folding	Niacin	[109,110,111]
**CMT4B2**	MTMR13	Knock-out mouse	MTMR13−/−	Mild	Demyelinating	Aberrant myelin folding		[112]
**CMT4C**	SH3TC2	Knock-out mouse	SH3TC2−/−	Mild	Demyelinating	Mild hypo/demyelination	Gene replacement therapy	[113,114,115]
**CMT4F**	PRX	Knock-out mouse	PRX−/−	Present	Demyelinating	Segmental demyelination		[116]
**CMT4J**	FIG4	Mutant mouse	Fig4−/− (Pale tremor mouse)	Present	Mixed	Severe hypo/demyelination and onion bulbs	AAV gene therapy	[117,118,119]
**CMTX**	CX32	Transgenic mouse	S26L (S3) hCX32	Present	N/A	N/A	CamKII inhibitor	[120,121]
**CMTX**	CX32	Transgenic mouse	G12S (G1,G2); S26L (S1,S2) hCX32	Mild	N/A	N/A	CamKII inhibitor	[120,121]
**CMTX**	CX32	Knock-out mouse	CX32−/−; CX32y/-	Mild	Mixed	Demyelination and onion bulbs	Intratecal gene delivery (early); AAV9-Mpz.GJB1 therapy; scAAV1. tMCK.NT-3	[122,123,124,125,126,127,128]
**CMT-DI**	EBP50	Knock-out mouse	EBP50+/−	Mild	Demyelinating	Mild axon atrophy		[129]
**CMT-DI**	C1orf194	Knock-out mouse	C1orf194−/+; C1orf194−/−	Present	Mixed	Demyelination		[130]
**CMT-DI**	C1orf194	Knock-in mouse	I121N/I121N; I121N/+	Present	Axonal	Demyelination		[131]
**CMTDIB**	DNM1	Knock-in mouse	DNM2 K552E/+	Mild	CMAP decrease (Myopatic)	Normal		[132]
**CMTDIC**	YARS	Transgenic mouse	AdhYARS E196K/ChAT mice	N/A	N/A	N/A		[133]
**ARCMT2**	TRIM2	Knock-out mouse	TRIM2A/A	Present	Axonal	Reduced axon number		[134]

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
