# Peer review of "Animal Models as a Tool to Design Therapeutical Strategies for CMT-like Hereditary Neuropathies"

_brainsci, 2021, doi:10.3390/brainsci11091237_

Round 1

Reviewer 1 Report

Overall impression

The authors present in this review the creation and the use of animal models for designing new therapeutic strategies for hereditary neuropathies. I find the subject very interesting and worth publication. The paper itself is very well written and easy to read even though it needs some major corrections, according to me.

Major corrections

Despite the fact that History and History of Medicine are very interesting fields, I find the 1st part (1. Introduction and historical use of animal models in medicine) way too long for an introduction to the subject. Ideally, it should be reduced to 1/3rd of the present length. I think, the removed sentences could become the start for an article of its own on history of animal models.

Similarly, the second part (2. Brief introduction to human Inherited Peripheral Neuropathies (IPNs)) is lengthy and not very informative. I would rather have seen a cellular and molecular description of the mechanisms involved, i.e. demyelination or axonal suffering and loss, without going too much in the details.

By the way, the title seems to broad since the article mainly focuses on CMT neuropathies when other inherited neuropathies exist with different pathophysiologies, such as the amyloid neuropathy linked to the transthyretin gene (TTR). Same for the Refsum neuropathy. So, I would suggest to slightly rephrase the title of the article to "Animal models as a tool to design therapeutical strategies for CMT-type hereditary neuropathies" or something like that.

The 3rd part (3. The importance of animal models in inherited neuropathies) is definitively the most important part of this article and contains a great load of information. However, I would have written it in a different way. I would have asked : are the current animal models for CMT really reliable, what are the prominent cellular and molecular features of the CMT neuropathies and how well are the animal models adressing these issues ? For instance, length-dependent neuropathy is a major point of these PN in humans. Is it possible to adress that point with small rodents, i.e. with peripheral nerves a few centimeters long ? Or, are the models adressing the axon-myelin sheath interactions (axonal loss after long term myelin suffering) ? I believe to increase even more the quality of this review this part should be revisited and improved.

I did not find the reference for the following article which is, I think, pertaining to the subject (Rodent models with expression of PMP22: Relevance to dysmyelinating CMT and HNPP. Jouaud M, Mathis S, Richard L, Lia AS, Magy L, Vallat JM. J Neurol Sci. 2019 Mar 15; 398:79-90. doi: 10.1016 /j.jns.2019.01.030. Epub 2019 Jan 21. PMID: 30685714 Review).

The 4th part is interesting as it shows the importance of these models for finding therapeutic means. Sadly, in the conventional drugs, curcumin is not mentionned in the list of the possible therapeutic means. I would suggest to fill this gap with this recent article "Curcumin-cyclodextrin/cellulose nanocrystals improve the phenotype of Charcot-Marie-Tooth-1A transgenic rats through the reduction of oxidative stress. Caillaud M, Msheik Z, Ndong-Ntoutoume GM, Vignaud L, Richard L, Favreau F, Faye PA, Sturtz F, Granet R, Vallat JM, Sol V, Desmoulière A, Billet F. Free Radic Biol Med. 2020 Dec;161:246-262. doi: 10.1016/j.freeradbiomed.2020.09.019. Epub 2020 Sep 25. PMID: 32980538".

Author Response

We thank the Reviewer for the positive comments and suggestions to ameliorate the manuscript. Please find below point-by-point response.

Overall impression

The authors present in this review the creation and the use of animal models for designing new therapeutic strategies for hereditary neuropathies. I find the subject very interesting and worth publication. The paper itself is very well written and easy to read even though it needs some major corrections, according to me.

Major corrections

Despite the fact that History and History of Medicine are very interesting fields, I find the 1st part (1. Introduction and historical use of animal models in medicine) way too long for an introduction to the subject. Ideally, it should be reduced to 1/3rd of the present length. I think, the removed sentences could become the start for an article of its own on history of animal models.

Answer: this part was widely reduced as suggested by the Reviewer. Further shrinkage would affect the comprehension of the paragraph. 

Similarly, the second part (2. Brief introduction to human Inherited Peripheral Neuropathies (IPNs)) is lengthy and not very informative. I would rather have seen a cellular and molecular description of the mechanisms involved, i.e. demyelination or axonal suffering and loss, without going too much in the details.

Answer: the manuscript is intended for general Reader, so we think it is necessary to give some general introduction to inherited neuropathies. However, we appreciated Reviewer’s considerations and following his/her suggestions, we shortened this general part and briefly introduced molecular mechanisms of demyelination and axonal degeneration (page 3).  

By the way, the title seems to broad since the article mainly focuses on CMT neuropathies when other inherited neuropathies exist with different pathophysiologies, such as the amyloid neuropathy linked to the transthyretin gene (TTR). Same for the Refsum neuropathy. So, I would suggest to slightly rephrase the title of the article to "Animal models as a tool to design therapeutical strategies for CMT-type hereditary neuropathies" or something like that.

Answer: we agree with the Reviewer and modified the title as suggested.

The 3rd part (3. The importance of animal models in inherited neuropathies) is definitively the most important part of this article and contains a great load of information. However, I would have written it in a different way. I would have asked : are the current animal models for CMT really reliable, what are the prominent cellular and molecular features of the CMT neuropathies and how well are the animal models adressing these issues ? For instance, length-dependent neuropathy is a major point of these PN in humans. Is it possible to adress that point with small rodents, i.e. with peripheral nerves a few centimeters long ? Or, are the models adressing the axon-myelin sheath interactions (axonal loss after long term myelin suffering) ? I believe to increase even more the quality of this review this part should be revisited and improved.

Answer: We thank the Reviewer for the advice. We better explicated those concepts already present in the manuscript and changed/included the other suggestions (from page 9 to 13).

I did not find the reference for the following article which is, I think, pertaining to the subject (Rodent models with expression of PMP22: Relevance to dysmyelinating CMT and HNPP. Jouaud M, Mathis S, Richard L, Lia AS, Magy L, Vallat JM. J Neurol Sci. 2019 Mar 15; 398:79-90. doi: 10.1016 /j.jns.2019.01.030. Epub 2019 Jan 21. PMID: 30685714 Review).

Answer: We thank you the Reviewer for the suggestion of this reference that we missed. It was included in the 3.1 section.

The 4th part is interesting as it shows the importance of these models for finding therapeutic means. Sadly, in the conventional drugs, curcumin is not mentionned in the list of the possible therapeutic means. I would suggest to fill this gap with this recent article "Curcumin-cyclodextrin/cellulose nanocrystals improve the phenotype of Charcot-Marie-Tooth-1A transgenic rats through the reduction of oxidative stress. Caillaud M, Msheik Z, Ndong-Ntoutoume GM, Vignaud L, Richard L, Favreau F, Faye PA, Sturtz F, Granet R, Vallat JM, Sol V, Desmoulière A, Billet F. Free Radic Biol Med. 2020 Dec;161:246-262. doi: 10.1016/j.freeradbiomed.2020.09.019. Epub 2020 Sep 25. PMID: 32980538".

Answer: we did not intend to treat all the therapies investigated for CMTs, as this argument would need a dedicated manuscript, but just to give some examples of what has been done / is on-going. We understand that curcumin is a major argument in the field and thus a paragraph on the use of curcumin has been included (page 14).

Reviewer 2 Report

Title: “Animal models as a tool to design therapeutical strategies for hereditary neuropathies”

Summary: The authors present their narrative review of the utility of animal models (emphasis in rodents) for inherited peripheral neuropathies. Overall, the manuscript it well written and informative and I have mostly minor comments (many grammatical) that should be addressed before recommending for publication. I will not list all of the grammatical changes that might be made, but only those that overtly disrupted my reading and appreciation for the science you present. Thus, I recommend a spelling/grammar check to fix the more minor errors.

  1. Line 81: Some minor grammatical edits here; add an "s" to "breakthrough" and remove the comma. In other words, this should be "These major breakthroughs in medical sciences progressively disproved..."
  2. Line 105: Another minor grammatical edit: "begun" should be "began."
  3. Line 105: Another minor grammatical edit: "begun" should be "began."
  4. Figure 1 legend: Capitalize "Use" as the first word in the sentence.
  5. Table 1 legend: Similar to Figure 1, capitalize the first word of the sentence; "Overview"

Note: the line numbering reset to 1 after the figure and table, hence the reset in my mention of line numbering as well.

  1. Section 3.1: "PMP22" is written in various different ways throughout the remainder of the manuscript. I.e., sometimes it is all caps and italicized (e.g., line 4), sometimes it is not italicized (e.g., line 7), and then sometimes only the first P is capitalized (e.g., line 10). Unless there is a systematic reason for the different formats, I recommend choosing one and being consistent throughout the manuscript. I have similar comments for MPZ, MTMR2, Fig4, HspB1, NEFL, NFM, and NFH. There may have been others that I missed, so a careful review of formatting consistency would be beneficial for the entire manuscript.
  2. Section 3.2, Line 70: "physical displace" should probably be "physical displacement"
  3. Section 3.2, Line 72: "where elicits" should be "where it elicits"
  4. Section 3.2, line 80: Here you use "Ser63del" instead of "S63del." Was this intentional?
  5. Section 3.3, lines 90-91: "In the previous paragraph, ..." It might be more appropriate to say "In the previous section, ..."
  6. Section 3.4, line 141: "Reduction" should not be capitalized.
  7. Section 4.2, line 243: "... (see paragraph 3.1)." I recommend replacing "paragraph" with "section," similar to my comment above.
  8. Section 4.2, line 273: I think you have duplicated the phrase "driven by" here.
  9. Author Contributions, lines 309-316: This is still the template provided by Brain Sciences. Please delete and use only the one you provide in lines 321-322.
  10. Funding, lines 317-320: It seems this is an erroneous template funding section as well. Please delete and use the funding statement on lines 323-324.

Author Response

We thank the Reviewer for positive comments ans suggestions. Find below point-by-point answers.

Title: “Animal models as a tool to design therapeutical strategies for hereditary neuropathies”

Summary: The authors present their narrative review of the utility of animal models (emphasis in rodents) for inherited peripheral neuropathies. Overall, the manuscript it well written and informative and I have mostly minor comments (many grammatical) that should be addressed before recommending for publication. I will not list all of the grammatical changes that might be made, but only those that overtly disrupted my reading and appreciation for the science you present. Thus, I recommend a spelling/grammar check to fix the more minor errors.

  1. Line 81: Some minor grammatical edits here; add an "s" to "breakthrough" and remove the comma. In other words, this should be "These major breakthroughs in medical sciences progressively disproved..."
  2. Line 105: Another minor grammatical edit: "begun" should be "began."
  3. Line 105: Another minor grammatical edit: "begun" should be "began."
  4. Figure 1 legend: Capitalize "Use" as the first word in the sentence.
  5. Table 1 legend: Similar to Figure 1, capitalize the first word of the sentence; "Overview"

Answer: this part has been fixed accordingly.

Note: the line numbering reset to 1 after the figure and table, hence the reset in my mention of line numbering as well.

  1. Section 3.1: "PMP22" is written in various different ways throughout the remainder of the manuscript. I.e., sometimes it is all caps and italicized (e.g., line 4), sometimes it is not italicized (e.g., line 7), and then sometimes only the first P is capitalized (e.g., line 10). Unless there is a systematic reason for the different formats, I recommend choosing one and being consistent throughout the manuscript. I have similar comments for MPZ, MTMR2, Fig4, HspB1, NEFL, NFM, and NFH. There may have been others that I missed, so a careful review of formatting consistency would be beneficial for the entire manuscript.

Answer: we used conventional rule writing the gene all capitalized when is referred to humans (or in case of general comment; i.e. MPZ) while it was capitalized only the first letter when referred to rodent gene (Mpz); if referred to the gene is in Italic (NEFL), while in normal plain test when referred to the protein (NEFL). We checked the text to correct inconsistencies.

  1. Section 3.2, Line 70: "physical displace" should probably be "physical displacement"
  2. Section 3.2, Line 72: "where elicits" should be "where it elicits"
  3. Section 3.2, line 80: Here you use "Ser63del" instead of "S63del." Was this intentional?
  4. Section 3.3, lines 90-91: "In the previous paragraph, ..." It might be more appropriate to say "In the previous section, ..."
  5. Section 3.4, line 141: "Reduction" should not be capitalized.
  6. Section 4.2, line 243: "... (see paragraph 3.1)." I recommend replacing "paragraph" with "section," similar to my comment above.
  7. Section 4.2, line 273: I think you have duplicated the phrase "driven by" here.

Answer: all these comments have been fixed accordingly.

  1. Author Contributions, lines 309-316: This is still the template provided by Brain Sciences. Please delete and use only the one you provide in lines 321-322.
  2. Funding, lines 317-320: It seems this is an erroneous template funding section as well. Please delete and use the funding statement on lines 323-324.

Answer: all these comments have been fixed accordingly.

Round 2

Reviewer 1 Report

This article has been noticeably improved. I recommend it for publication.